# Deanonymization in the Bitcoin P2P Network

**Giulia Fanti and Pramod Viswanath**

## Abstract

Recent attacks on Bitcoin's peer-to-peer (P2P) network demonstrated that its transaction-flooding protocols, which are used to ensure network consistency, may enable user deanonymization—the linkage of a user's IP address with her pseudonym in the Bitcoin network. In 2015, the Bitcoin community responded to these attacks by changing the network's flooding mechanism to a different protocol, known as diffusion. However, it is unclear if diffusion actually improves the system's anonymity. In this paper, we model the Bitcoin networking stack and analyze its anonymity properties, both pre- and post-2015. The core problem is one of epidemic source inference over graphs, where the observational model and spreading mechanisms are informed by Bitcoin's implementation; notably, these models have not been studied in the epidemic source detection literature before. We identify and analyze near-optimal source estimators. This analysis suggests that Bitcoin's networking protocols (both pre- and post-2015) offer poor anonymity properties on networks with a regular-tree topology. We confirm this claim in simulation on a 2015 snapshot of the real Bitcoin P2P network topology.

## 1 Introduction

The Bitcoin cryptocurrency has seen widespread adoption, due in part to its reputation as a privacy-preserving financial system [17, 22]. In practice, though, Bitcoin exhibits serious privacy vulnerabilities [3, 19, 27, 28, 24]. Most of these vulnerabilities arise because of two key properties: (1) Bitcoin associates each user with a pseudonym, and (2) pseudonyms can be linked to financial transactions through a public transaction ledger, called the *blockchain* [23]. If an attacker can associate a pseudonym with a human identity, the attacker may learn the user's transaction history.

In practice, there are several ways to link a user to her Bitcoin pseudonym. The most commonly-studied methods analyze transaction patterns in the public blockchain, and link those patterns using side information [3, 19, 27, 28, 24]. In this paper, we are interested in a lower-layer vulnerability: the networking stack. Like most cryptocurrencies, Bitcoin nodes communicate over a P2P network [23]. Whenever a user (Alice) generates a transaction (i.e., sends bitcoins to another user, Bob), she first creates a "transaction message" that contains her pseudonym, Bob's pseudonym, and the transaction amount. Alice subsequently floods this transaction message over the P2P network, which enables other users to validate her transaction and incorporate it into the global blockchain.

The anonymity implications of transaction broadcasting were largely ignored until recently, when researchers demonstrated practical deanonymization attacks on the P2P network [6, 15]. These attacks use a "supernode" to connect to all active Bitcoin nodes and listen to the transaction traffic they relay [15, 6, 7]. By using simple estimators to infer the source IP of each transaction broadcast, this *eavesdropper adversary* was able to link IP addresses to Bitcoin pseudonyms with an accuracy of up to 30% [6]. We refer to such linkage as *deanonymization*.

Giulia Fanti (`gfanti@andrew.cmu.edu`) is in the ECE Department at Carnegie Mellon University. Pramod Viswanath (`pramodv@illinois.edu`) is in the ECE Department at the University of Illinois at Urbana-Champaign.

This work was funded by NSF grant CCF-1705007.

In 2015, the Bitcoin community responded to these attacks by changing its flooding protocols from a gossip-style protocol known as *trickle spreading* to a *diffusion spreading* protocol that spreads content with independent exponential delays [1]. We define these protocols precisely in Section 2. However, no systematic motivation was provided for this shift. Indeed, it is unclear whether the change actually defends against the deanonymization attacks in [6, 15].

**Problem and contributions.** The main point of our paper is to show that Bitcoin's flooding protocols have poor anonymity properties, and the community's shift from trickle spreading (pre-2015) to diffusion spreading (post-2015) did not help the situation. The problem of deanonymizing a user in this context is mathematically equivalent to inferring the source of a random spreading process over a graph, given partial observations of the spread. The optimal (maximum-likelihood) source-identification algorithms change between spreading protocols; identifying such algorithms and quantifying their accuracy is the primary focus of this work. We find that despite having different maximum-likelihood estimators, trickle and diffusion exhibit roughly the same, poor anonymity properties. Our specific contributions are threefold:

*(1) Modeling.* We model the Bitcoin P2P network and an eavesdropper adversary, whose capabilities reflect recent practical attacks in [6, 15]. Most Bitcoin network protocols are not explicitly documented, so modeling the system requires parsing a combination of documentation, papers, and code. Several of the resulting models are new to the epidemic source detection literature.

*(2) Analysis of Trickle (Pre-2015).* We analyze the probability of deanonymization by an eavesdropper adversary under trickle propagation. Our analysis is conducted over a regular tree-structured network. Although the Bitcoin network topology is not a regular tree, we show in Section 2 that regular trees are a reasonable first-order model. We consider graph-independent estimators (e.g., the first-timestamp estimator), as well as maximum-likelihood estimators; both are defined precisely in Section 2. Our analysis suggests that although the first-timestamp estimator performs poorly on high-degree trees, maximum-likelihood estimators achieve high probabilities of detection for trees of any degree $d$.

*(3) Analysis of Diffusion (Post-2015).* We conduct a similar analysis of diffusion spreading, which was adopted in 2015 as a fix for the anonymity weaknesses observed under trickle propagation [6, 15]. The analysis of diffusion requires different theoretical tools, including nonlinear differential equations and generalized Pòlya urns. Although the analysis techniques and attack mechanisms are different, we find that the anonymity properties of diffusion are similar to those of trickle. Namely, the first-timestamp estimator's probability of detection decays to 0 as degree $d$ grows, but the maximum-likelihood probability of detection remains high (in particular, non-vanishing) even as $d \to \infty$.

## 2 Model and related work

**Network model.** We model the P2P network of Bitcoin nodes as a graph $G(V, E)$, where $V$ is the set of all server nodes and $E$ is the set of edges, or connections, between them. Each server is represented by a (IP address, port) tuple; it can establish up to eight outgoing connections to other Bitcoin nodes [6, 2]. The resulting sparse random graph between nodes can be modeled approximately as a 16-regular graph; in practice, the average degree is closer to 8 due to nonhomogeneities across nodes [20]. The graph is locally tree-like and (approximately) regular. For this reason, *regular trees* are a natural class of graphs to study. In our theoretical analysis, we model $G$ as a $d$-regular tree. We validate this choice by running simulations on a snapshot of the true Bitcoin network [20] (Section 5).

**Spreading protocols.** Each transaction must be broadcast over the network; we analyze the spread of a single message originating from source node $v^* \in V$. Without loss of generality, we label $v^*$ as node '0' when iterating over nodes. At time $t = 0$, the message starts spreading according to one of two randomized protocols: trickle (pre-2015) or diffusion (post-2015).

*Trickle spreading* is a gossip-based flooding protocol. Each source or relay chooses a neighboring peer (called the 'trickle' node) uniformly at random, every 200 ms. If the trickle node has not yet received the message, the sender forwards the message [6].[1] We model this by considering a canonical, simpler spreading protocol of *round-robin gossip*. In round-robin gossip, each source or relay randomly orders its neighbors who have not yet seen the message; we call these *uninfected* neighbors. In each successive (discrete) timestep, the node transmits the message to the next neighbor

in its ordering. Thus, if a node has $d$ neighbors, all $d$ neighbors will receive the message within $d$ timesteps. This differs from trickle spreading, where the time-to-infection is a coupon collector's problem, and therefore takes $\Theta(d \log d)$ timesteps in expectation [8]. We will henceforth abuse terminology by referring to round-robin gossip as trickle spreading.

In *diffusion*, each source or relay node transmits the message to each of its uninfected neighbors with an independent, exponential delay of rate $\lambda$. In practice, Bitcoin uses a higher rate on outgoing edges than incoming ones [2]; we omit this distinction in our model. We assume a continuous-time system, with each node starting the exponential clocks upon receipt (or creation) of a message.

For both protocols, we let $X_v$ denote the timestamp at which node $v \in V$ receives a given message. Note that server nodes cannot be infected more than once. We assume the message originates at time $t = 0$, so $X_{v^*} = X_0 = 0$. Moreover, we let $G_t(V_t, E_t)$ denote the *infected subgraph* of $G$ at time $t$, or the subgraph of nodes who have received the message (but not necessarily reported it to the adversary) by time $t$.

**Adversarial model.** The adversary's goal is to link a message with the source (IP address, port)—i.e., to identify the source node $v^* \in V$. We consider an *eavesdropper adversary*, whose capabilities are modeled on the practical deanonymization attacks in [6, 15]. These attacks use a supernode that connects to most of the servers in the Bitcoin network. It can make multiple connections to each honest server, with each connection coming from a different (IP address, port). Hence, the honest server does not realize that the supernode's connections are all from the same entity. We model this by assuming that the eavesdropper adversary makes a fixed number $\theta$ of connections to each server, where $\theta \geq 1$. We do not include these adversarial connections in the original server graph $G$, so $G$ remains a $d$-regular graph (see Figure 1). The supernode can learn the network structure between servers [6], so we assume that $G(V, E)$ is known to the eavesdropper.

The supernode in [6, 15] observes the timestamps at which messages are relayed from each honest server, without relaying or transmitting content. If the adversary maintains multiple active connections to each server ($\theta > 1$), it receives the message $\theta$ times from each server. We let $\tau_v$ denote the time at which the adversary *first* observes the message from node $v \in V$. We let $\boldsymbol{\tau} = (\tau_v)_{v \in V}$ denote the set of all observed first-timestamps. We assume timestamps are relative to time $t = 0$, i.e., the adversary knows when the message started spreading.

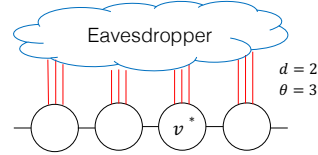

Figure 1: The eavesdropper adversary establishes $\theta$ links (in red) to each server. Honest servers are connected in a $d$-regular tree topology (edges in black).

**Source estimation.** The adversary's goal is as follows: given the observed timestamps $\boldsymbol{\tau}$ (up to estimation time $t$) and the graph $G$, find an estimator $\mathtt{M}(\boldsymbol{\tau}, G)$ that outputs the true source. Our metric of success for the adversary is *probability of detection*, $\mathbb{P}(\mathtt{M}(\boldsymbol{\tau}, G) = v^*)$, taken over the random spreading realization (captured by $\boldsymbol{\tau}$) and any randomness in the estimator.

In [6, 15], the adversary uses a variant of the *first-timestamp estimator* $\mathtt{M}_{\mathtt{FT}}(\boldsymbol{\tau}, G) = \arg\min_{v \in V_t} \tau_v$, which outputs the first node (prior to estimation time $t$) to report the message to the adversary. The first-timestamp estimator requires no knowledge of the graph, and it is computationally easy to implement. We begin by analyzing this estimator for both trickle and diffusion propagation.

We also consider the *maximum-likelihood (ML) estimator*: $\mathtt{M}_{\mathtt{ML}}(\boldsymbol{\tau}, G) = \arg\max_{v \in V} \mathbb{P}(\boldsymbol{\tau} | G, v^* = v)$. The ML estimator depends on the time of estimation $t$ to the extent that $\boldsymbol{\tau}$ only contains timestamps up to time $t$. Unlike the first-timestamp estimator, the ML estimator differs across spreading protocols, depends on the graph, and may be computationally intractable in general.

**Problem statement.** Our goal is to understand whether the Bitcoin community's move from trickle spreading to diffusion actually improved the system's anonymity guarantees. The problem at hand is to characterize the maximum-likelihood (ML) probability of detection of the eavesdropper adversary for both trickle and diffusion processes on $d$-regular trees, as a function of degree $d$, number of corrupted connections $\theta$, and detection time $t$. We meet this goal by computing lower bounds derived from the analysis of suboptimal estimators (e.g., first-timestamp estimator and centrality-based estimators), and upper bounds derived from fundamental limits on detection.

**Related work.** Although there has been much work on the anonymity properties of Bitcoin [19, 28, 24, 27], the 'epidemic source finding' interpretation of Bitcoin deanonymization is fairly new. Prior work that (implicitly) adopts this interpretation has focused on Bitcoin's protocol flaws more than the inference aspect of the problem [6, 15]. As this is the focus of our paper, we include the related source detection literature. Epidemic source detection has been widely studied under diffusion spreading with a *snapshot adversary*, which observes the set of infected nodes at a single time $t$; in our notation, the adversary would learn the set $\{v \in V : X_v \leq t\}$ (no timestamps), along with graph $G$. Shah and Zaman first characterized the ML probability of detection for diffusion observed by a snapshot adversary when the underlying graph is a regular tree [29]. These results were later extended to random, irregular trees [31], whereas other authors studied heuristic source detection methods on general graphs [12, 26, 16] and related theoretical limits [32, 21, 14]. The eavesdropper adversary differs in that it eventually observes a noisy timestamp $\tau_v$ from *every* node, regardless of when the node is infected. This changes both the analysis and the estimators that one can use. Another common adversarial model is the *spy-based adversary*, which observes exact timestamps for a corrupted set of nodes that does not include the source [25, 34]. In our notation, for a set of spies $S \subseteq V$, the spy-based adversary observes $\{(s, X_s) : s \in S\}$. Prior work on the spy-based adversary does not characterize the ML probability of detection, but researchers have proposed efficient heuristics that perform well in practice [25, 34, 35, 9]. Unlike the spy-based adversary, the eavesdropper only observes delayed timestamps, and it does so for *all* nodes, including the source.

# 3 Analysis of trickle (pre-2015)

## 3.1 First-timestamp estimator

The analysis of trickle propagation is complicated by its combinatorial, time-dependent nature. As such, we lower-bound the first-timestamp estimator's probability of detection. Let $\tau_m \triangleq \min(\tau_1, \tau_2, \ldots)$ denote the minimum observed timestamp among nodes that are *not* the source. Then we compute $\mathbb{P}(\tau_0 < \tau_m)$, i.e., the probability that the true source reports the message to the adversary strictly before any of the other nodes. This event (which causes the source to be detected with probability 1) does not include cases where the true source is one of $k$ nodes ($k > 1$) that report the message to the adversary simultaneously, and before any other node in the system. Nonetheless, for large node degree $d$, the 'simultaneous reporting' event is rare, so our lower bound is close to the empirical probability of detection of the first-timestamp estimator.

**Theorem 3.1** *(Proof in Appendix C.1) Consider a message that propagates according to trickle spreading over a $d$-regular tree of servers, where each node additionally has $\theta$ connections to an eavesdropping adversary. The first-timestamp estimator's probability of detection at time $t = \infty$ satisfies* $\mathbb{P}(M_{FT}(\boldsymbol{\tau}, G) = v^*) \geq \frac{\theta}{d \log 2} \left[ Ei(2^d \log \rho) - Ei(\log \rho) \right]$ *where* $\rho = \frac{d-1}{d-1+\theta}$, *and* $Ei(x) \triangleq -\int_{-x}^{\infty} \frac{e^{-t} dt}{t}$ *denotes the exponential integral.*

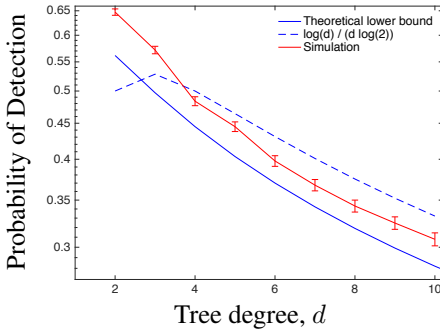

Figure 2: First-timestamp estimator accuracy on $d$-regular trees when $\theta = 1$.

We prove this bound by conditioning on the time at which the source reports to the adversary. The proof then becomes a combinatorial counting problem. The expression in Theorem 3.1 can be simplified by examining its Taylor expansion (see Appendix A). In particular, for the special case of $\theta = 1$ where the adversary establishes only one connection per server, line (5) simplifies to $\mathbb{P}(M_{\mathrm{FT}}(\boldsymbol{\tau}, G)) \approx \frac{\log d}{d \cdot \log 2} + o\left(\frac{\log d}{d}\right)$. This suggests that the first-timestamp estimator has a probability of detection that decays to zero asymptotically as $\log(d)/d$. Intuitively, the probability of detection should decay to zero, because the higher the degree of the tree, the higher the likelihood that a node *other* than the source reports to the adversary before the source does. Nonetheless, this is only a lower bound on the first-timestamp's probability of detection, so we wish to understand how tight the bound is.

**Simulation.** To evaluate the lower bound in Theorem 3.1 and its approximation for $\theta = 1$, we simulate the first-timestamp estimator on regular trees.[2] Figure 2 illustrates the simulation results for $\theta = 1$ compared to the approximation above. Each data point is averaged over 5,000 trials. Empirically, the lower bound appears to be tight, especially as $d$ grows. Figure 2 suggest a natural solution to improve anonymity in the Bitcoin network: increase the degree of each node to reduce the adversary's probability of detection. However, we shall see in the next section that stronger estimators (e.g., the ML estimator) may achieve high probabilities of detection, even for large $d$.

## 3.2 Maximum-likelihood estimator

At any time $t$, if one knew the ground truth timestamps (i.e., the $X_v$'s), one could arrange the nodes of the infected subgraph $G_t$ in the order they received the message. We call such an arrangement an *ordering* of nodes. Since propagation is in discrete time, multiple nodes may receive the message simultaneously; such nodes are lumped together in the ordering. Of course, the true ordering is not observed by the adversary, but the observed timestamps (i.e., $\boldsymbol{\tau}$) restrict the set of possible orderings. A *feasible ordering* is an ordering that respects the rules of trickle propagation over graph $G$, as well as the observed timestamps $\boldsymbol{\tau}$. In this subsection only, we will abuse notation by using $\boldsymbol{\tau}$ to refer to *all* timestamps observed by the adversary, not just the first timestamp from each server. So if the adversary has $\theta$ connections to each server, $\boldsymbol{\tau}$ would include $\theta$ timestamps per honest server.

We propose an estimator called **timestamp rumor centrality**, which counts the number of feasible orderings originating from each candidate source. The candidate with the most feasible orderings is chosen as the estimator output. This estimator is similar to rumor centrality, an estimator devised for snapshot adversaries in [29]. However, the presence of timestamps and the lack of knowledge of the infected subgraph increases the estimator's complexity. We first motivate timestamp rumor centrality.

**Proposition 3.2** *(Proof in Appendix C.2) Consider a trickle process over a d-regular graph, where each node has $\theta$ connections to the eavesdropper adversary. Any feasible orderings $o_1$ and $o_2$ with respect to observed timestamps $\boldsymbol{\tau}$ and graph $G$ have the same likelihood.*

Proposition 3.2 implies that at any fixed time, the likelihood of observing $\boldsymbol{\tau}$ given a candidate source is proportional to the number of feasible orderings originating from that candidate source. Therefore, an ML estimator (timestamp rumor centrality) counts the number of feasible orderings at estimation time $t$. Timestamp rumor centrality is a message-passing algorithm that proceeds as follows: for each candidate source, recursively determine the set of feasible times when each node could have been infected, given the observed timestamps. This is achieved by passing a set of "feasible times of receipt" from the candidate source to the leaves of the largest feasible infected subtree rooted at the candidate source. In each step, nodes prune receipt times that conflict with their observed timestamps. Next, given each node's set of feasible receipt times, they count the number of feasible orderings that obey the rules of trickle propagation. This is achieved by passing sets of partial orderings from the leaves to the candidate source, and pruning infeasible orderings. The timestamp rumor centrality protocol is presented in Appendix A.2, along with minor modifications that reduce its complexity.

In [31], precise analysis of standard rumor centrality was possible because rumor centrality can be reduced to a simple counting problem. Such an analysis is more challenging for timestamp rumor centrality, because timestamps prevent us from using the same counting argument. However, we identify a suboptimal, simplified version of timestamp rumor centrality that approaches optimal probabilities of detection as $t$ grows. We call this estimator **ball centrality**.

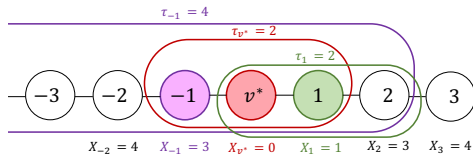

Figure 3: Example of ball centrality on a line with one link to the adversary per server (these links are not shown). The estimator is run at time $t = 4$.

Ball centrality checks whether a candidate source $v$ could have generated each of the observed timestamps, *independently*. For example, Figure 3 contains a sample spread on a line graph, where the adversary has one connection per server (not shown). Therefore, $d = 2$ and $\theta = 1$. The ground truth infection time is written as $X_v$ below each node, and the observed timestamps are written

[2]Code for all simulations available at `https://github.com/gfanti/bitcoin-trickle-diffusion`.

above the node. In this figure, the estimator is run at time $t = 4$, so the adversary only sees three timestamps. For each observed timestamp $\tau_v$, the estimator creates a ball of radius $\tau_v - 1$, centered at $v$. For example, in our figure, the green node (node 1) has $\tau_1 = 2$. Therefore, the adversary would make a ball of radius 1 centered at node 1; this ball is depicted by the green bubble in our figure. The ball represents the set of nodes that are close enough to node 1 to feasibly report to the adversary from node 1 at time $\tau_1 = 2$. After constructing an analogous ball for every observed timestamp in $\boldsymbol{\tau}$, the protocol outputs a source selected uniformly from the intersection of these balls. In our example, there are exactly two nodes in this intersection. We describe ball centrality precisely in Protocol 1 (Appendix A.2.1). Although ball centrality is not ML for a fixed time $t$, the following theorem lower bounds the ML probability of detection by analyzing ball centrality and showing that its probability of detection approaches a fundamental upper bound exponentially fast in detection time $t$.

**Theorem 3.3** *(Proof in Section C.3) Consider a trickle spreading process over a $d$-regular graph of honest servers. In addition, each server has $\theta$ independent connections to an eavesdropper adversary. The ML probability of detection at time $t$ satisfies the following expression:*

$$1 - \frac{d}{2(\theta + d)} - \left( \frac{d}{\theta + d} \right)^t \overset{(a)}{\leq} \mathbb{P}(M_{ML}(\boldsymbol{\tau}, G) = v^*) \overset{(b)}{\leq} 1 - \frac{d}{2(\theta + d)} \quad (1)$$

Note that the right-hand side of equation (1) is always greater than $\frac{1}{2}$. As such, increasing the graph degree would not significantly reduce the probability of detection; the adversary can still identify the source with probability at least $\frac{1}{2}$ given enough time. Second, the ML probability of detection approaches its upper bound exponentially fast in time $t$. This suggests that the adversary can achieve high probabilities of detection at small times $t$. These results highlight an important point: estimators that exploit graph structure can reap significant, order-level gains in accuracy.

## 4 Analysis of diffusion (post-2015)

### 4.1 First-timestamp estimator

Although the first-timestamp estimator does not use knowledge of the underlying graph, its performance depends on the underlying graph structure. The following theorem exactly characterizes its probability of detection on a regular tree as $t \to \infty$.

**Theorem 4.1** *(Proof in Appendix C.4) Consider a diffusion process of rate $\lambda = 1$ over a $d$-regular tree, $d > 2$. Suppose an adversary observes each node's infection time with an independent, exponential delay of rate $\lambda_2 = \theta$, $\theta \geq 1$. Then the following expression describes the probability of detection for the first-timestamp estimator at time $t = \infty$: $\mathbb{P}(M_{FT}(\boldsymbol{\tau}, G) = v^*) = \frac{\theta}{d-2} \log \left( \frac{d+\theta-2}{\theta} \right).$*

The proof expresses the probability of detection as a nonlinear differential equation that can be solved exactly. The expression highlights a few points: First, for a fixed degree $d$, the probability of detection is strictly positive as $t \to \infty$. This is straightforward to see, but under other adversarial models (e.g., snapshot adversaries) it is not trivial to see that the probability of detection is positive as $t \to \infty$. Indeed, several papers are dedicated to making that point [30, 31]. Second, when $\theta = 1$, i.e., the adversary has only one connection per node, the probability of detection approaches $\log(d)/d$ asymptotically in $d$. This quantity tends to 0 as $d \to \infty$, and it is order-equal to the probability of detection of the first-timestamp adversary on the trickle protocol when $\theta = 1$ (see Section 3.1).

Theorem 4.1 suggests that the Bitcoin community's transition from trickle spreading to diffusion does not provide order-level anonymity gains (asymptotically in the degree of the graph), at least for the first-timestamp adversary. Next, we ask if the same is true for estimators that use the graph structure.

### 4.2 Centrality-based estimators

We compute a different lower bound on the ML probability of detection by analyzing a centrality-based estimator. Unlike the first-timestamp estimator, this *reporting centrality estimator* uses the structure of the infected subgraph by selecting a candidate source that is close to the center (on the graph) of the observed timestamps. However, it does not explicitly use the observed timestamps. Also unlike the first-timestamp estimator, this centrality-based estimator improves as the degree $d$

of the underlying tree increases, with a strictly positive probability of detection as $d \to \infty$. Thus the eavesdropper adversary has an ML probability of detection that scales as $\Theta(1)$ in $d$. Intuitively, reporting centrality works as follows: for each candidate source $v$, the estimator counts the number of nodes that have reported to the adversary from each of the node $v$'s adjacent subtrees. It picks a candidate source for which the number of reporting nodes is approximately equal in each subtree.

To make this precise, suppose the infected subtree $G_t$ is rooted at $w$; we use $T_v^w$ to denote the subtree of $G_t$ that contains $v$ and all of $v$'s descendants, with respect to root node $w$. Consider a random variable $Y_v(t)$, which is 1 if node $v \in V$ has reported to the adversary by time $t$, and 0 otherwise. We let $Y_{T_v^w}(t) = \sum_{u \in T_v^w} Y_u(t)$ denote the number of nodes in $T_v^w$ that have reported to the adversary by time $t$. We use $Y(t) = \sum_{v \in V_t} Y_v(t)$ to denote the total number of reporting nodes in $G_t$ at time $t$. Similarly, we use $N_{T_v^w}(t)$ to denote the number of *infected* nodes in $T_v^w$ (so $N_{T_v^w}(t) \geq Y_{T_v^w}(t)$), and we let $N(t)$ denote the total number of infected nodes at time $t$ ($N(t) \geq Y(t)$). For each candidate source $v$, we consider its $d$ neighbors, which comprise the set $\mathcal{N}(v)$. We define a node $v$'s *reporting centrality* at time $t$—denoted $R_v(t)$—as follows:

$$R_v(t) = \begin{cases} 1 & \text{if } \max_{u \in \mathcal{N}(v)} Y_{T_u^v}(t) < \frac{Y(t)}{2} \\ 0 & \text{otherwise.} \end{cases} \tag{2}$$

That is, a node's reporting centrality is 1 iff each of its adjacent subtrees has fewer than $Y(t)/2$ reporting nodes. A node is a *reporting center* iff its reporting centrality is 1. The estimator outputs $\hat{v}$ chosen uniformly from all reporting centers. In Figure 4, $v^*$ is the only reporting center.

Reporting centrality does not use the adversary's observed timestamps—it only counts the number of reporting nodes in each of a node's adjacent subtrees. This estimator is inspired by *rumor centrality* [30], an ML estimator for the source of a diffusion process under a snapshot adversary. Recall that a snapshot adversary sees the infected subgraph $G_t$ at time $t$, but it does not learn timestamp information.

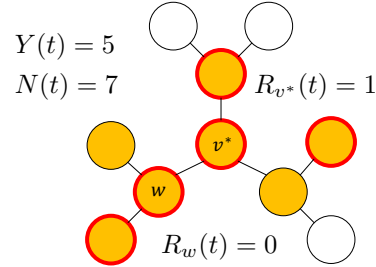

$Y(t) = 5$
$N(t) = 7$
$R_{v^*}(t) = 1$
$R_w(t) = 0$

The next theorem shows that for trees with high degree $d$, reporting centrality has a strictly higher (in an order sense) probability of detection than the first-timestamp estimator; its probability of detection is strictly positive as $d \to \infty$.

Figure 4: Yellow nodes are infected; a red outline means the node has reported. $R_{v^*}(t) = 1$ since $v^*$'s adjacent subtrees have $\leq Y(t)/2 = 2.5$ reporting nodes.

**Theorem 4.2** *(Proof in Section C.5) Consider a diffusion process of rate $\lambda = 1$ over a $d$-regular tree. Suppose this process is observed by an eavesdropper adversary, which sees each node's timestamp with an independent exponential delay of rate $\lambda_2 = \theta$, $\theta \geq 1$. Then the reporting centrality estimator has a (time-dependent) probability of detection $\mathbb{P}(M_{RC}(\tau, G) = v^*)$ that satisfies $\liminf_{t \to \infty} \mathbb{P}(M_{RC}(\tau, G) = v^*) \geq C_d > 0$. where $C_d = 1 - d\left(1 - I_{1/2}\left(\frac{1}{d-2}, 1 + \frac{1}{d-2}\right)\right)$ is a constant that depends only on degree $d$, and $I_{1/2}(a,b)$ is the regularized incomplete Beta function, i.e., the probability a Beta random variable with parameters $a$ and $b$ takes a value in $[0, \frac{1}{2})$.*

To prove this, we relate two Pòlya urn processes: one that represents the diffusion process over the regular tree of honest nodes, and one that describes the full spreading process, which includes both diffusion over the regular tree and random reporting to the adversary. The first urn can be posed as a classic Pòlya urn [10], which has been studied in the context of diffusion [31, 14]. The second urn can be described by an unbalanced generalized Pòlya urn (GPU) with negative coefficients [4, 13]—a class of urns that does not typically appear in the study of diffusion (to the best of our knowledge). As a side note, this approach can be used to analyze other epidemic source-finding problems that have previously evaded analysis, as we show in Appendix B. Notice that the constant $C_d$ in Theorem 4.2 does not depend on $\theta$—this is because the reporting centrality estimator makes no use of timestamp information, so the delays in the timestamps $\tau$ do not affect the estimator's asymptotic behavior.

**Simulation results**. To evaluate the lower bound in Theorem 4.2, we simulate reporting centrality on diffusion over regular trees. Figure 5 illustrates the empirical performance of reporting centrality averaged over 4,000 trials, compared to the theoretical lower bound on the liminf. The estimator is

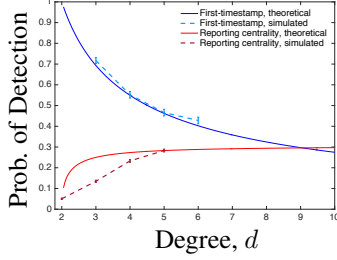
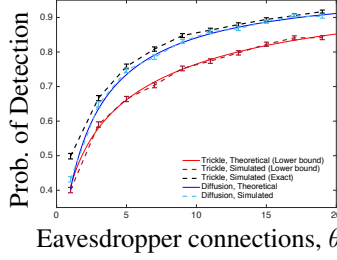
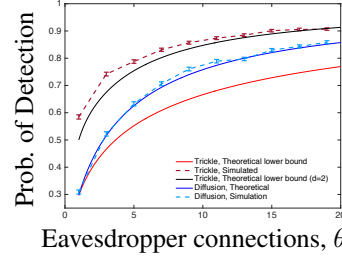

Figure 5: First-timestamp vs. reporting centrality on diffusion over regular trees, theoretically and simulated. $\theta = 1, t = d + 2$.

Figure 6: Comparison of trickle and diffusion under the first-timestamp estimator on 4-regular trees.

Figure 7: Trickle vs. diffusion under the first-timestamp estimator, simulated on a snapshot of the real Bitcoin network [20].

Table 1: Probability of detection on a $d$-regular tree. The adversary has $\theta$ connections per server.

| | | Trickle | Diffusion |
|---|---|---|---|
| **First-Timestamp** | All $\theta$ | $\frac{\theta\left[\mathrm{Ei}(2^d \log \rho) - \mathrm{Ei}(\log \rho)\right]}{d \log 2}$ (Thm 3.1) | $\frac{\theta}{d-2}\log\left(\frac{d+\theta-2}{\theta}\right)$ (Thm. 4.1) |
| | $\theta = 1$ | $\frac{\log(d)}{d\log(2)} + o\left(\frac{\log d}{d}\right)$ (Sec. 3.1) | $\frac{\log(d-1)}{(d-2)}$ (Thm. 4.1) |
| **Maximum-Likelihood** | All $\theta$ | $1 - \frac{d}{2(\theta+d)}$ (Thm 3.3) | $1 - d\left(1 - I_{1/2}\left(\frac{1}{d-2}, 1 + \frac{1}{d-2}\right)\right)$ |
| | $\theta = 1$ | $1 - \frac{d}{2(d+1)}$ (Thm 3.3) | (Thm. 4.2) |

run at time $t = d + 2$. Our simulations are run up to degree $d = 5$ due to computational constraints, since the infected subgraph grows exponentially in the degree of the tree. By $d = 5$, reporting centrality reaches the theoretical lower bound on the limiting detection probability.

For diffusion, neither lower bound on the first-timestamp or reporting centrality estimator strictly outperforms the other. Figure 5 compares the two estimators as a function of degree $d$. We observe that reporting centrality outstrips first-timestamp estimation for trees of degree 9 and higher; since our theoretical result is only a lower bound on the performance of reporting centrality, the transition may occur at even smaller $d$. Empirically, the true Bitcoin graph is approximately 8-regular [20], a regime in which we expect reporting centrality to perform similarly to the first-timestamp estimator.

## 5  Discussion

Table 1 summarizes our theoretical results for trickle and diffusion. The probabilities of detection for trickle and diffusion are similar, particularly when $\theta = 1$. Although the maximum-likelihood results are difficult to compare visually, they both approach a positive constant as $d, t \to \infty$; for trickle propagation, that constant is $\frac{1}{2}$, whereas for diffusion, it is approximately 0.307.

These results are asymptotic in degree $d$. In practice, the underlying Bitcoin graph is fixed; the only variable quantity is the adversary's resources, represented by $\theta$. Figure 6 compares analytical expressions and simulations for 4-regular trees under the first-timestamp estimator (as we lack an ML estimator on general graphs), as a function of $\theta$. It suggests nearly identical detection probabilities for diffusion and trickle on regular trees; while our theoretical prediction for diffusion is exact, our lower bound on trickle is loose since $d$ is small.

To validate our decision to analyze regular trees, we simulate trickle and diffusion on a 2015 snapshot of the Bitcoin network [20]. Figure 7 compares these results as a function of $\theta$, for the first-timestamp estimator. Unless specified otherwise, theoretical curves are calculated for a regular tree with $d = 8$, the mean degree of our dataset. Diffusion performs close to the theoretical prediction; this is because with high probability, the first-timestamp estimator uses only on a local neighborhood to estimate $v^*$, and the Bitcoin graph is locally tree-like. However, our trickle lower bound remains loose. This is partially due to simultaneous reporting events, but the main contributing factor seems to be graph irregularity. Understanding this effect more carefully is an interesting question for future work.

In summary, trickle and diffusion have similar probabilities of detection, both in an asymptotic-order sense and numerically. We have analyzed the canonical class of $d$-regular trees and simulated these protocols on a real Bitcoin graph topology. Our results omit certain details of the spreading protocols, (Sec. 2); extending the analysis to include these details is practically relevant.

## Footnotes

[1]This description omits some details of trickle spreading, which we do not consider in our analysis. For example, with probability 1/4, each relay forwards the message instantaneously to its neighbors *without* trickling.

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
