[Supplementary Material]

# Supplementary material

## A    Trickle discussion and algorithms

### A.1    First-timestamp estimator

We repeat Theorem 3.1 for ease of reading:

**Theorem A.1** *Consider a message that propagates according to trickle spreading over a $d$-regular tree of servers, where each node additionally has $\theta$ connections to an eavesdropping adversary. The first-timestamp estimator's probability of detection at time $t = \infty$ satisfies $\mathbb{P}(M_{FT}(\boldsymbol{\tau}, G) = v^*) \geq \frac{\theta}{d \log 2} \left[ Ei(2^d \log \rho) - Ei(\log \rho) \right]$ where $\rho = \frac{d-1}{d-1+\theta}$, and $Ei(x) \triangleq - \int_{-x}^{\infty} \frac{e^{-t} dt}{t}$ denotes the exponential integral.*

We can approximate the asymptotic behavior of this theorem for large $d$ by using the exponential integral's Taylor expansion. First, we note that when $d$ is large, $\text{Ei}(2^d \log \rho) \approx 0$, so we have

$$\frac{\theta}{d \log 2} \left[ \text{Ei}(2^d \log \rho) - \text{Ei}(\log \rho) \right] \approx \frac{\theta}{d \log 2} \left( -\gamma - \log|\log \rho| - \sum_{\nu=1}^{\infty} \frac{(\log \rho)^\nu}{\nu \cdot \nu!} \right) \tag{3}$$

$$\approx \frac{\theta}{d \log 2} \left( -\gamma - \log \log \left( 1 + \frac{\theta}{d} \right) + \log(1 + \frac{\theta}{d}) \quad - \frac{\log^2(1 + \frac{\theta}{d})}{4} + \dots \right) \tag{4}$$

$$\approx \frac{\theta}{d \log 2} \left( -\gamma - \log \frac{\theta}{d} + \frac{\theta}{d} - \frac{\theta^2}{4d^2} + \dots \right) \tag{5}$$

where $\gamma \approx 0.577$ is the Euler-Mascheroni constant [33], and (3) comes from substituting the exponential integral by its Taylor expansion for real arguments [5]. Line (4) holds because as $d \to \infty$, $\rho \approx \frac{1}{1+\theta/d}$, and (5) holds because as $d \to \infty$, $\log(1 + \frac{\theta}{d}) \approx \frac{\theta}{d}$.

### A.2    Maximum-likelihood estimator: timestamp rumor centrality

As described in Section 3.2, timestamp rumor centrality is a message-passing algorithm. Leaves of the candidate infected subgraph pass partial orderings of the nodes' infection times back to the root (i.e., the candidate source). In practice, we do not pass the entire partial ordering; we can instead store the *number* of distinct partial orderings for the subtree rooted at each child, indexed by each feasible time of receipt for the child (there are $O(d)$ feasible times of receipt). This reduces the overall computational complexity to $O((2d)^d |V|)$: each node passes a message of size $O(d)$ to its parent, and each parent takes a Cartesian product of its children's messages. The whole procedure is run for each candidate source, of which there are $O(2^d)$, since the true source has a timestamp of at most $d + 1$. The full protocol is given in Figure 9. Figure 9 assumes that the adversary has only one link per honest server, so $\theta = 1$. This is easily extended to general $\theta$.

In practice, timestamp rumor centrality seems to achieve a fundamental upper bound on probability of detection. Figure 8 shows the probability of detection of timestamp rumor centrality, as a function of $d$, for $\theta = 1$. The estimator is run at time $t = d+1$. Even at such small timestamps, the probability of detection is empirically close to the upper bound in (1).

#### A.2.1    Ball centrality

The full protocol for ball centrality is included in Protocol 1 below.

**Protocol 1** BALL CENTRALITY. Returns a source estimate whose location is consistent with timestamps $\boldsymbol{\tau}$ on tree $G$. $h(v, w)$ denotes the hop distance between $v$ and $w$.

**Input:** Timestamps $\boldsymbol{\tau}$, graph $G(V, E)$
**Output:** Source estimate $\hat{v} \in V$
1: $W \leftarrow V$
2: **for** $v \in V$ **do**                                          ▷ Find the intersection of feasible balls
3:      $W \leftarrow W \cap \{w \in V : h(w, v) \leq \tau_w - 1\}$
4: $\hat{v} \sim \text{Unif}(W)$

## B   Diffusion discussion and algorithms

We wish to highlight the connection between reporting centrality and *rumor centrality* [30]. Rumor centrality is an ML estimator for determining the source of a diffusion process under a snapshot adversary. Recall that a snapshot adversary gets to see which nodes are infected at estimation time $t$, but it does not see any timestamps. Rumor centrality counts the number of feasible orderings for each candidate source that could have generated the observed snapshot. Since [30] considers a continuous-time system with exponential delays, rumor centrality does need to account for concurrent events.

Rumor centrality exhibits a key property on trees: a node $v$ is a rumor center of a tree-structured infected subgraph $G_t$ if and only if each of the $d$ adjacent subtrees adjacent has no more than $N(t)/2$ nodes in it [30]:

$$\max_{u \in \mathcal{N}(v)} N_{T_u^v}(t) \leq \frac{N(t)}{2}.$$

Moreover, there exists at least one and at most two rumor centers in a tree, and the true source has a strictly positive probability of being a rumor center on regular trees and geometric random trees [30]. In our case, we cannot use rumor centrality directly because the true infected nodes are not all observed at any point in time $t$. We construct the reporting centrality estimator by applying a condition like the one for rumor centrality to the *reporting* nodes, rather than the infected nodes.

The techniques used to analyze diffusion under an eavesdropper adversary can also be used to study the spy-based adversary. The ML probability of detection for the spy-based adversary on regular trees has evaded exact analysis [11], though several heuristics have been found to work well [25, 34]. A simple lower bound on the ML probability of detection comes from analyzing the first-timestamp estimator for the spy-based adversary [11]. This gives

$$\liminf_{t \to \infty} \mathbb{P}(\texttt{M}_{\texttt{FT}}(\boldsymbol{\tau}, G) = v^*) \geq p, \tag{6}$$

Figure 8: Timestamp rumor centrality (ML estimator) accuracy on $d$-regular trees when the adversary has one link per server ($\theta = 1$). The estimator is run at time $t = d + 1$.

Figure 9: TIMESTAMP RUMOR CENTRALITY. Returns the timestamp rumor centrality of candidate source $v$, given timestamps $\tau$ on tree $G$. $\phi(w)$ denotes the children of node $w$ on a tree $T$ rooted at $v$. $\partial T$ denotes the leaves of tree $T$.

**Input:** Timestamps $\tau$, graph $G$, candidate source $v$, estimation time $t \geq d + 1$
**Output:** Number feasible orderings from source $v$, $f_v$
1: $[d + 1] \triangleq \{1, \dots, d + 1\}$
2: $m_{vv} \leftarrow \{0\}$
3:        ▷ Two global variables, $T$ and $\mathcal{C}$:
4: $T \leftarrow$ balanced tree of radius $t$, center $v$, over graph $G$
5: $\mathcal{C} \leftarrow$ dictionary of number of feasible partial orderings for each feasible $(w, \hat{X}_w)$ pair
6: $c = \text{PASSTOLEAVES}(v, v, \{0\}, T)$
7: **return** $c$ ▷ Timestamp rumor centrality for $v$
8:
9: **function** PASSTOLEAVES$(z, w, m, T)$   ▷ Each node learns its possible receipt times, recursively
10:     $m(w) \leftarrow m \setminus \{t \in m : t \geq \tau_w\}$   ▷ Remove late times
11:     Node $w$ saves $m(w)$ outside function scope
12:     **if** $w \in \partial T$ **then**
13:         **for** $t' \in m$ **do**
14:             $\mathcal{C}[w][t'] \leftarrow 1$ ▷ Number of partial orderings with $w$ infected at time $t'$
15:     **else**
16:         $m' \leftarrow \cup_{i \in m(w)} \{i + j : j \in [d + 1]\}$
17:         $m' \leftarrow m' \setminus \{\tau_w\}$
18:         **for** $y \in \phi(w)$ **do**
19:             PASSTOLEAVES$(w, y, m')$
20:     AGGREGATEMESSAGES$(w, \phi(w))$

21:     **if** $z = w = v^*$ **then**
22:         **return** $\mathcal{C}[v^*][0]$
23:
24: **function** AGGREGATEMESSAGES$(z, \phi(z), T)$ ▷ Counts the number of valid orderings by passing messages that represent the set of feasible orderings
25:     $\mathcal{N} \leftarrow [z, \phi(z)]$   ▷ Ordered list of parent and children
26:     $M \leftarrow \prod_{u \in \phi(z)} m(u)$
27:         ▷ $\prod$ denotes Cartesian set product, where $A \times B \triangleq \{(a, b) : a \in A, b \in B\}$
28:     $M \leftarrow m(w) \times M$         ▷ Prepends the current node's feasible receipt times to the Cartesian product
29:     $M \leftarrow \{m \in M : [(m_1, \dots, m_d) \text{ distinct}] \wedge [|m_i - m_j| \leq d + 1 \text{ for all } i, j] \wedge [m_1 < m_i \text{ for all } i > 1]\}$
30:            ▷ Removes ordered tuples where neighbors receive the message at the same time, are too far apart to be feasible, or parent gets message after children. $m_i$ denotes $i$th element of ordered tuple $m$
31:     **for** m $\in M$ **do**
32:         $q \leftarrow \prod_{i=2}^{d} \mathcal{C}[\mathcal{N}_i][m_i]$   ▷ Compute the number of permutations by multiplying counts from each child node
33:         $\mathcal{C}[z][m_1] \leftarrow C[z][m_1] + q$
34:     **return**

since with probability $p$, the first node to receive the message is a spy. Although reporting centrality is a suboptimal source estimator for the eavesdropper adversary, it is straightforward to analyze and can be used to obtain lower bounds on the probability of detection for the spy-based adversary.

Using reporting centrality and similar proof techniques to Theorem 4.2, we can lower-bound the spy-based adversary's ML probability of detection as time $t \to \infty$.

**Corollary B.1** *Consider a diffusion process of rate $\lambda = 1$ over a $d$-regular tree. Suppose this process is observed by an spy-based adversary, which sees the exact timestamp of each node independently with probability $p > 0$ (otherwise it sees nothing). Then the reporting centrality estimator has a probability of detection $\mathbb{P}(M_{RC}(\tau, G) = v^*)$ that satisfies*

$$\liminf_{t \to \infty} \mathbb{P}(M_{RC}(\tau, G) = v^*) \geq C_d > 0, \tag{7}$$

*where $C_d = 1 - d \left( 1 - I_{1/2} \left( \frac{1}{d-2}, 1 + \frac{1}{d-2} \right) \right)$. (Proof in Section C.6)*

The key point to notice about Corollary B.1 is that (7) remains positive for a fixed $d$ even as $p \to 0$; this is in contrast with the previously-known equation (6). Keep in mind that we are first taking the limit as $t \to \infty$ for a fixed $p$, then taking $p \to 0$. It is not possible to exchange the order of these limits; doing so drives the probability of detection to zero. The reason is that for a finite $t$ (and hence a finite number of infected nodes), as $p \to 0$, the number of spies in the infected subgraph also tends to

zero. Without any observations, it is impossible to detect the source. However, for any fixed positive $p > 0$, as $t \to \infty$, eventually the number of spies in each subtree concentrates, so reporting centrality has a nonzero probability of detection. Indeed, our analysis makes critical use of the fact that as $t \to \infty$, for a fixed positive $p$, the classical Pòlya urn we use to represent the underlying diffusion process concentrates.

Although the lower bound in (7) does not depend on $p$, the convergence rate does; understanding this convergence rate is of theoretical interest. More broadly, a deeper understanding of the similarities between the eavesdropper adversary and other canonical adversarial models is needed.

## C  Proofs

### C.1  Proof of theorem 3.1

We can write this lower bound explicitly. Recall that the system is discrete-time, and each node has $d$ honest neighbors and $\theta$ connections to the adversary. $\tau_i$ denotes the *first* time node $i$ reports to the adversary.

$$\mathbb{P}(\tau_0 < \tau_m) = \sum_{i=1}^{d+\theta} \mathbb{P}(\tau_0 = i)\mathbb{P}(\tau_m > i|\tau_0 = i) \tag{8}$$

Since the source has only $d$ honest connections, for any $i > (d+1)$, it holds that $\mathbb{P}(\tau_0 = i) = 0$, and if $i = d+1$, then $\tau_m \leq i$. So we can simplify this summation to

$$\mathbb{P}(\tau_0 < \tau_m) = \sum_{i=1}^{d} \mathbb{P}(\tau_0 = i)\mathbb{P}(\tau_m > i|\tau_0 = i). \tag{9}$$

This means we only need to consider the first $d$ time steps of the message spread in order to lower-bound the first-timestamp adversary's probability of detection. Let $a_j \triangleq \mathbb{P}(\tau_m = j|\tau_0 \geq j) = \mathbb{P}(\tau_m = j|\tau_0 = i$ for any $i \geq j)$. Then $\mathbb{P}(\tau_m > i|\tau_0 = i) = 1 - \sum_{j=1}^{i} a_j$, so

$$\mathbb{P}(\tau_0 < \tau_m) = \qquad \sum_{i=1}^{d} \mathbb{P}(\tau_0 = i) - \sum_{i=1}^{d} \mathbb{P}(\tau_0 = i)\sum_{j=1}^{i} a_j$$

$$= \qquad 1 - \mathbb{P}(\tau_0 = d+1) - \sum_{i=1}^{d} \mathbb{P}(\tau_0 = i)\sum_{j=1}^{i} a_j. \tag{10}$$

Letting $b_k \triangleq \mathbb{P}(\tau_m > k \mid \tau_0 \geq k, \tau_m > (k-1))$ gives

$$a_j = (1 - b_j)\prod_{k=1}^{j-1} b_k. \tag{11}$$

We can write $b_k$ explicitly by noting that as long as no infected nodes have reported the message to the adversary, one can deterministically compute the number of infected nodes at each time step with a given number of infected (resp. uninfected) neighbors. This is because the underlying graph is a regular tree (see Lemma 3.2 for proof). Let $\mathcal{N}(k,t)$ denote the set of nodes with $k$ infected, honest neighbors at time $t$. Then for a fixed time $t$, we can compute the probability that *every* infected node chooses to infect an honest node in the next time step, by indexing over the value of $k$:

$$b_j = \prod_{k=1}^{j-1} \left( \frac{|\{v \in V : u_{v,h}(j) = k\}|}{|\{v \in V : u_v(j) = k\}|} \right)^{|\mathcal{N}(k,j)|}$$

$$= \prod_{k=1}^{j-1} \left( \frac{d-k}{d-k+\theta} \right)^{2^{j-1-k}}$$

where $j > 1$, and $b_1 = 1$. Here, the quantity $u_v(j)$ (resp. $u_{v,h}(j)$) denotes the uninfected degree (resp. uninfected honest degree) of node $v$ at time $j$—that is, the number of total (resp. honest),

uninfected neighbors of a node. So the ratio in the definition of $b_j$ is comparing the number of nodes with honest uninfected degree $k$ to the number of nodes with uninfected degree $k$.

Substituting this quantity into Equation (11), we get

$$
\begin{aligned}
a_j &= (1-b_j)\prod_{k=1}^{j-2}\prod_{m=1}^{k}\left(\frac{d-m}{d-m+\theta}\right)^{2^{k-m}} \\
&= (1-b_j)\prod_{m=1}^{j-2}\left(\frac{d-m}{d-m+\theta}\right)^{\sum_{\ell=0}^{j-m-2}2^{\ell}} \\
&= (1-b_j)\underbrace{\prod_{m=1}^{j-2}\left(\frac{d-m}{d-m+\theta}\right)^{2^{j-m-1}-1}}_{M_j}
\end{aligned}
\tag{12}
$$

Rearranging, we get

$$
\begin{aligned}
M_j &= \frac{\prod_{m=1}^{j-1}\left(\frac{d-m}{d-m+\theta}\right)^{2^{j-1-m}}\prod_{k=1}^{j-2}\left(\frac{d-k+\theta}{d-k}\right)}{\frac{d-j+1}{d+\theta-j+1}} \\
&= b_j\left(\frac{d+\theta-j+1}{d-j+1}\right)\underbrace{\prod_{k=1}^{j-2}\left(\frac{d-k+\theta}{d-k}\right)}_{W_j}
\end{aligned}
$$

Writing out the terms of $W_j$ explicitly, we get that

$$
\begin{aligned}
W_j &= \frac{(d-1+\theta)(d-2+\theta)\dots(d-1)\dots(d-j+2+\theta)}{(d-1)\dots(d-j+2+\theta)\dots(d-j+2)} \\
&= \prod_{m=0}^{\theta-1}\frac{d+m}{d+m-j+2}.
\end{aligned}
$$

Substituting all of this into Equation (12), we get that

$$
a_j = (1-b_j)b_j\left(\frac{d-j+1+\theta}{d-j+1}\right)\prod_{m=0}^{\theta-1}\frac{d+m}{d+m-j+2}.
$$

This expression for $a_j$ can be used to rewrite (10):

$$
\begin{aligned}
\mathbb{P}(\tau_0<\tau_m) &= 1-\underbrace{\mathbb{P}(\tau_0=d+1)}_{P_A}-\sum_{i=1}^{d}\mathbb{P}(\tau_0=i)\sum_{j=1}^{i}a_j. \\
&= 1-P_A-\sum_{i=1}^{d}\mathbb{P}(\tau_0=i)\sum_{j=1}^{i}(1-b_j)b_j\times \\
&\qquad \frac{d-j+1+\theta}{d-j+1}\prod_{m=0}^{\theta-1}\frac{d+m}{d+m-j+2}.
\end{aligned}
\tag{13}
$$

Setting

$$
q_j \triangleq \frac{d-j+1+\theta}{d-j+1}\prod_{m=0}^{\theta-1}\frac{d+m}{d+m-j+2},
$$

we get that

$$
\mathbb{P}(\tau_0<\tau_m) = \qquad\qquad 1-P_A-\sum_{i=1}^{d}\mathbb{P}(\tau_0=i)\sum_{j=1}^{i}(1-b_j)b_jq_j
$$

$$
= \qquad\qquad 1-P_A-\sum_{i=1}^{d}\mathbb{P}(\tau_0=i)\left(\sum_{j=2}^{i}b_jq_j-\sum_{j=2}^{i}b_j^2q_j\right)
\tag{14}
$$

where the change of summation bounds in (14) occurs because $b_1 = 1$. We define $\gamma_k \triangleq \left(\frac{d-k}{d-k+\theta}\right)^{2^{-k}}$, which means that

$$b_j = \prod_{k=1}^{j-1} \gamma_k^{2^{j-1}}. \tag{15}$$

Using this notation, we write out the two final summations in (14) explicitly:

$$\sum_{j=2}^{i} b_j q_j = q_2 \gamma_1^2 + q_3 \gamma_1^4 \gamma_2^4 + q_4 \gamma_1^8 \gamma_2^8 \gamma_3^8 + \ldots + q_i \gamma_1^{2^{i-1}} \cdots \gamma_{i-1}^{2^{i-1}} \tag{16}$$

$$\sum_{j=2}^{i} b_j^2 q_j = q_2 \gamma_1^4 + q_3 \gamma_1^8 \gamma_2^8 + q_4 \gamma_1^{16} \gamma_2^{16} \gamma_3^{16} + \ldots + q_i \gamma_1^{2^i} \cdots \gamma_{i-1}^{2^i} \tag{17}$$

Subtracting (16)-(17) and collecting terms gives

$$\sum_{j=2}^{i} q_j (b_j - b_j^2) = q_2 \gamma_1^2 + \gamma_1^4 (q_3 \gamma_2^4 - q_2) + \gamma_1^8 \gamma_2^8 (q_4 \gamma_3^8 - q_3) +$$

$$\ldots + \gamma_1^{2^{i-1}} \cdots \gamma_{i-2}^{2^{i-1}} (q_i \gamma_{i-1}^{2^{i-1}} - q_{i-1}) - q_i \gamma_1^{2^i} \cdots \gamma_{i-1}^{2^i}$$

$$= q_2 \gamma_1^2 - q_i \gamma_1^{2^i} \cdots \gamma_{i-1}^{2^i} + \sum_{\ell=3}^{i} \gamma_1^{2^{\ell-1}} \cdots \gamma_{\ell-2}^{2^{\ell-1}} (q_\ell \gamma_{\ell-1}^{2^{\ell-1}} - q_{\ell-1}). \tag{18}$$

First, we show that the summation (last term) in (18) is equal to 0 by writing out $(q_\ell \gamma_{\ell-1}^{2^{\ell-1}} - q_{\ell-1})$:

$$q_\ell \gamma_{\ell-1}^{2^{\ell-1}} - q_{\ell-1} = q_{\ell-1} \left( \frac{d-\ell+1+\theta}{d-\ell+1} \gamma_{\ell-1}^{2^{\ell-1}} - 1 \right)$$

$$= q_{\ell-1} \left( \frac{d-\ell+1+\theta}{d-\ell+1} \cdot \frac{d-\ell+1}{d-\ell+1+\theta} - 1 \right) = 0.$$

Next, we show that the first term in (18) equals 1 by writing $q_2 \gamma_1^2$ explicitly:

$$q_2 \gamma_1^2 = \frac{d-2+1+\theta}{d-2+1} \left( \prod_{m=0}^{\theta-1} \frac{d+m}{d+m-2+2} \right) \frac{d-1}{d-1+\theta} = 1,$$

which implies that

$$\sum_{j=1}^{i} q_j (b_j - b_j^2) = 1 - q_i \gamma_1^{2^i} \cdots \gamma_{i-1}^{2^i} = 1 - b_i^2. \tag{19}$$

Now, we can substitute (19) into (14), getting

$$\mathbb{P}(\tau_0 < \tau_m) = 1 - P_A - \sum_{i=2}^{d} \mathbb{P}(\tau_0 = i)(1 - q_i \gamma_1^{2^i} \cdots \gamma_{i-1}^{2^i})$$

$$= \frac{\theta}{\theta+d} + \sum_{i=2}^{d} \mathbb{P}(\tau_0 = i) q_i \gamma_1^{2^i} \cdots \gamma_{i-1}^{2^i}, \tag{20}$$

where (20) is because

$$\sum_{i=2}^{d} \mathbb{P}(\tau_0 = i) = 1 - \mathbb{P}(\tau_0 = d+1) - \mathbb{P}(\tau_0 = 1)$$

$$= 1 - P_A - \frac{\theta}{\theta+d}.$$

Note that $P(\tau_0 = i) = \frac{\binom{N-i}{\theta-1}}{\binom{N}{\theta}}$, where $N \triangleq d + \theta$. We can also rewrite $q_i$ as

$$q_i = \frac{d - i + 1 + \theta}{d - i + 1} \prod_{m=0}^{\theta-1} \frac{d + m}{d + m - i + 2}$$

$$= \frac{d - i + 1 + \theta}{d - i + 1} \cdot \frac{(d + \theta - 1)!}{(d-1)!} \cdot \frac{(d - i + 1)!}{(d + \theta + 1 - i)!} \cdot \frac{\theta!}{\theta!}$$

$$= \frac{\binom{N-1}{\theta}}{\binom{N-i}{\theta}}.$$

Thus, the product of these terms is

$$P(\tau_0 = i)q_i = \frac{\theta}{N - i - \theta + 1} \cdot \frac{N - \theta}{N}$$

$$= \frac{\theta}{d - i + 1} \cdot \frac{d}{\theta + d},$$

and the desired probability in (20) simplifies to

$$\mathbb{P}(\tau_0 < \tau_m) = \frac{\theta}{\theta + d} + \sum_{i=2}^{d} \frac{d}{d - i + 1} \cdot \frac{\theta}{\theta + d} b_i^2$$

$$= \frac{\theta}{\theta + d} \left[ 1 + d \sum_{i=2}^{d} \frac{b_i^2}{d - i + 1} \right]$$

$$\geq \frac{\theta}{\theta + d} \left[ 1 + \sum_{i=2}^{d} b_i^2 \right] \tag{21}$$

$$\geq \frac{\theta}{\theta + d} \left[ 1 + \sum_{i=1}^{d-1} \gamma_1^{\sum_{k=1}^{i} 2^k} \right] \tag{22}$$

$$= \frac{\theta}{\theta + d} \left[ 1 + \sum_{i=1}^{d-1} \gamma_1^{2^{i+1} - 2} \right]$$

$$\geq \frac{\theta}{d} \sum_{i=0}^{d-1} \gamma_1^{2^{i+1}} \tag{23}$$

where (21) comes from replacing $d - i + 1$ with $d$, (22) comes from replacing $\gamma_k$ with $\gamma_1$ since $\gamma_k \geq \gamma_1$, and (23) holds because $\gamma_1^2 \leq \frac{d}{d+\theta}$. We lower-bound the doubly exponential sum in (23) by integrating. Letting $\rho = \gamma_1^2$, we have

$$\frac{\theta}{d} \sum_{i=0}^{d-1} (\gamma_1^2)^{2^i} \geq \frac{\theta}{d} \int_0^d \rho^{2^x} dx$$

$$= \frac{\theta}{d \log 2} \left( \mathrm{Ei}(2^d \log \rho^2) - \mathrm{Ei}(\log \rho^2) \right)$$

where $\mathrm{Ei}(\cdot)$ denotes the exponential integral. This gives the desired result.

## C.2 Proof of proposition 3.2

The proposition can be seen through a simple counting argument. Let $A_t$ denote the set of *active nodes* at time $t$, or the set of all infected, honest nodes with at least one uninfected neighbor (honest or adversarial). We also define the *uninfected degree* of a node $u_v(t)$ as the number of uninfected neighbors of node $v$ at time $t$. We define $A_t(i) = |\{v \in V : u_v(t) = i\}|$, $i > 0$, as the number of active nodes at time $t$ with uninfected degree $i$.

We claim that for a given regular tree $G$ and set of observed timestamps $\tau$, and for any set of feasible orderings, the number of nodes with uninfected degree $i$ (i.e., $A_t(i)$) for a given $i > 0$ does not

depend on the underlying ordering. This holds because the graph is regular; we can show it formally by induction. At time $t = 1$, there are two options: either the source reports directly to the adversary, or it spreads the message to an honest neighbor. It if reports to the adversary, then in every feasible ordering, the source must report to the adversary at $t = 1$. Therefore at the end of time step $t = 1$, we have

$$A_t(i) = \begin{cases} 1 & \text{if } i = d + \theta - 1 \\ 0 & \text{otherwise.} \end{cases}$$

If the source instead passes the message to an honest neighbor (it doesn't matter which one, and the identity of the neighbor could vary across feasible orderings), then

$$A_t(i) = \begin{cases} 2 & \text{if } i = d + \theta - 1 \\ 0 & \text{otherwise,} \end{cases}$$

because now we have two active nodes, each of which has $d + \theta - 1$ uninfected neighbors.

Now take $t > 1$, and assume that at time $t - 1$, $A_t(i)$ was the same across all feasible orderings, for each $i > 0$. We want to show that the same is true at time $t$. Every time an active node $v$ infects a neighbor node $w$, $v$'s own uninfected degree decreases by one. If $w$ is an honest node, it joins the set of active nodes with uninfected degree $u_w(t) = d - 1 + \theta$. If $w$ belongs to the eavesdropper, then it does not join the active nodes. Suppose $\tau$ indicates that in time $t$, exactly $m$ nodes will report to the adversary. Since there were $|A_{t-1}|$ active nodes at time $t$, we know that $|A_{t-1}| - m$ new active nodes will be infected, each with degree $d - 1 + \theta$, so $A_t(d - 1 + \theta) = |A_{t-1}| - m$. Moreover, for all $0 < i < (d - 1 + \theta)$, we have $A_t(i) = A_{t-1}(i + 1)$, since each previously-active node decrements its uninfected degree by one. None of this depends on the ordering, which proves that $A_t(i)$ takes the same value for any feasible ordering.

We write out the likelihood $L(o)$ of a feasible ordering $o$:

$$L(o) = \prod_{j=1}^{t-1} \prod_{v \in A_j} \frac{1}{u_v(j)},$$

since each active node infects exactly one uninfected neighbor uniformly at random in each time step. By the previous argument, this likelihood can equivalently be grouped by nodes with the same uninfected degree $u$:

$$L(o) = \prod_{j=1}^{t} \prod_{u \in \{1, \ldots, d-1+\theta\}} \left( \frac{1}{u} \right)^{A_j(u)}.$$

Nothing in this expression depends on the ordering $o$ (since $A_j(u)$ is independent of $o$), so the likelihood must be equal for all feasible orderings.

## C.3 Proof of theorem 3.3

We begin by showing that (1) is an upper bound on any estimator's probability of detection, and then show that ball centrality achieves the lower bound in (1).

*(1):* Notice that with probability $\theta/(\theta + d)$, the true source reports the message to the adversary in the first time slot. If this happens, then the conditional probability of detection is 1; since the adversary is assumed to know the starting time of the trickle process, it can deduce that the true source is the only possible source. However, if the source does *not* report to the adversary in the first time slot, then the source must instead pass the message to one of its honest neighbors. In that case, at $t = 1$, exactly two nodes are infected, both of which are honest. Thereafter, the spread from each of these nodes is completely symmetric: both the true source and the first neighbor are connected to identical graph structures (i.e., two infinite trees rooted at each of the infected nodes), and the message spreading dynamics from each of these two nodes is identically-distributed. As such, no estimator can distinguish between the true source and the first neighbor, meaning the probability of detection is upper bounded by $1/2$ in this case. Therefore, a simple upper bound for the ML probability of detection is

$$\mathbb{P}(\mathbb{M}_{\mathrm{ML}}(\boldsymbol{\tau}, G) = v^*) \leq \frac{\theta}{\theta + d}(1) + \frac{d}{\theta + d}\left( \frac{1}{2} \right)$$

$$= 1 - \frac{d}{2(\theta + d)}.$$

*(2):* Next, we demonstrate that ball centrality has a probability of detection that is lower bounded by (1). Once again, if the source immediately passes the message to the adversary, then the adversary identifies a ball of size 1, so the source gets caught with probability 1. This happens with probability $\frac{\theta}{\theta+d}$. Thus we need to compute the probability of detection conditioned on the source passing the message to an honest neighbor at $t = 1$. Suppose source $v^*$ passes the message to its neighbor $w$ at $t = 1$. Without loss of generality, let us think of $w$ and $v^*$ as the left and right respective endpoints of a path. In each subsequent time step, each of the endpoints of this path will forward the message either to another honest node or to the adversary. If the left (resp. right) endpoint $q$ forwards the message to another honest node $z$, then $z$ becomes the left (resp. right) endpoint in the following time step; if $q$ forwards the message to the adversary, then the path terminates at $q$. This path continues to grow until both ends have terminated.

We first show that the probability of the path terminating is lower bounded by $1 - \frac{2}{(1+\theta/d)^{t-1}}$, then we show that conditioned on path termination, the probability of detection is lower bounded by $1/2$. To analyze the probability of termination at time $t$, note that each end of the path terminates after a geometrically-distributed number of hops. This holds because each endpoint independently terminates with probability $\frac{\theta}{\theta+d-1}$ in each time step. Therefore, the probability of both endpoints terminating by time $t$ can be expressed as

$$
\begin{aligned}
\mathbb{P}(B \le t)^2 &= \left(1 - \left(1 - \frac{\theta}{\theta+d-1}\right)^{t-1}\right)^2, \\
&\ge \left(1 - \left(\frac{d}{\theta+d}\right)^{t-1}\right)^2 \\
&\ge 1 - 2\left(\frac{d}{\theta+d}\right)^{t-1}
\end{aligned}
\tag{24}
$$

where $B$ is a geometric random variable of rate $\frac{\theta}{\theta+d-1}$.

Now, we show that conditioned on termination, the probability of detection is lower-bounded by $\frac{1}{2}$. We call $x$ and $y$ left and right terminating endpoints, respectively (Fig. 10).

Figure 10: Arrangement of nodes from the proof of Thm. 3.3.

Now, we show that the balls centered at nodes $x$ and $y$ have an intersection of at most two nodes. The ball centered at node $x$ has a radius of $\tau_x - 1$, and the ball centered at $y$ a radius of $\tau_y - 1$. By construction, we know that the hop distance between $v^*$ and $x$ is $h(v^*, x) = \tau_x - 1 \le \tau_x - 1$ and $h(v^*, y) = \tau_y - 2 \le \tau_y - 1$, so $v^*$ must lie in the intersection of the two balls. Similarly, $h(w, x) = \tau_x - 2 \le \tau_x - 1$ and $h(w, y) = \tau_y - 1 \le \tau_y - 1$, so $w$ must lie in the intersection of the two balls. Let us assume by contradiction that there exists a third node $z$ in the intersection of these two balls. This implies that $h(z, x) \le \tau_x - 1$ and $h(z, y) \le \tau_y - 1$. Either $z$ lies on the shortest path between $x$ and $y$, which we denote $P(x, y)$, or it does not. If $z \in P(x, y)$, then it either lies to the left of $w$ or to the right of $v^*$. In either case, $z$ is excluded from the ball centered at $y$ or $x$, respectively. Thus it cannot lie on $P(x, y)$. If $z$ does *not* lie on $P(x, y)$, then there exists an alternative path $P'(x, y) \ne P(x, y)$ between $x$ and $y$ of distance at most $\tau_x + \tau_y - 2$ that contains node $z$; this path is allowed to traverse the same node or edge multiple times. By construction, the hop distance between $x$ and $y$ is $h(x, y) = \tau_x + \tau_y - 3$, so $P'(x, y)$ must have at least this many hops. Moreover, since $G$ is a tree, every path between $x$ and $y$ must traverse each node in $P(x, y)$. Since $P(x, y)$ already contains $\tau_x + \tau_y - 3$ edges, $P'(x, y)$ should be no longer than $\tau_x + \tau_y - 2$, and it should also touch an additional node $z$, $P'(x, y)$ must have exactly one more hop than $P(x, y)$. This eliminates all paths that move from node $r \in P(x, y)$ to node $z \notin P(x, y)$, then back to $r$ again. But this implies that there exist two distinct paths between $x$ and $y$ in which no edge or vertex is traversed twice, which is a contradiction since $G$ is a tree. Hence there can be at most two nodes in the intersection of balls, so the probability of detection is at least 1/2 in this case.

$$\mathbb{P}(\tau_0 < \tau_m)$$

$$= \int_{t_0=0}^{\infty} \mathbb{P}(\tau_0 = t_0) \left[ \mathbb{P}(R_1 > t_0) + \int_{r_1=0}^{t_0} \mathbb{P}(R_1 = r_1)\mathbb{P}(\tau_1 > (t_0 - r_1)) \times \right.$$

$$\left. \left[ \mathbb{P}(R_2 > (t_0 - r_1)) + \int_{r_2=0}^{t_0-r_1} \mathbb{P}(R_2 = r_2)\mathbb{P}(\tau_2 > (t_0 - r_1 - r_2)) \left[ \mathbb{P}(R_3 > (t_0 - r_1 - r_2)) + \ldots \right]^{d-1} \right]^{d-1} \right]^{d_0} dt_0$$

$$= \int_0^{\infty} \lambda_2 e^{-\lambda_2 t_0} \left[ e^{-\lambda_1 t_0} + \int_0^{t_0} \lambda_1 e^{-\lambda_1 r_1} e^{-\lambda_2(t_0-r_1)} \left[ e^{-\lambda_1(t_0-r_1)} + \int_0^{t_0-r_1} \lambda_1 e^{-\lambda_1 r_2} e^{-\lambda_2(t_0-r_1-r_2)} \times \right. \right.$$

$$\left. \left. \left[ e^{-\lambda_1(t_0-r_1-r_2)} + \ldots \right]^{d-1} dr_2 \right]^{d-1} dr_1 \right]^{d_0} dt_0$$

$$= \int_0^{\infty} \lambda_2 e^{-\lambda_2 t_0} \left[ e^{-t_0} + \int_0^{t_0} e^{-r_1} e^{-\lambda_2(t_0-r_1)} \left[ e^{-(t_0-r_1)} + \int_0^{t_0-r_1} e^{-r_2} e^{-\lambda_2(t_0-r_1-r_2)} \times \right. \right.$$

$$\left. \left. \left[ e^{-(t_0-r_1-r_2)} + \ldots \right]^{d-1} dr_2 \right]^{d-1} dr_1 \right]^{d_0} dt_0$$

$$= \int_0^{\infty} \lambda_2 e^{-t_0(\lambda_2+d_0)} \left[ 1 + e^{-t_0(\lambda_2+d-2)} \int_0^{t_0} e^{r_1(\lambda_2+d-2)} \left[ 1 + e^{-(t_0-r_1)(\lambda_2+d-2)} \int_0^{t_0-r_1} e^{r_2(\lambda_2+d-2)} \times \right. \right.$$

$$\left. \left. \left[ 1 + \ldots \right]^{d-1} dr_2 \right]^{d-1} dr_1 \right]^{d_0} dt_0 \tag{25}$$

Figure 11: Probability of detection of the first-timestamp estimator, where $d$ is the node degree. $d_0$ denotes the degree of the source, which we eventually set to $d_0 = d - 1$ for symmetry. We condition on the source's reporting time $\tau_0 = t_0$. $R_i$ denotes the delay of infection times between $i$'s parent and $i$.

Combining this with the lower bound in (24), we have

$$\mathbb{P}(\mathtt{M_{BC}}(\boldsymbol{\tau}, G) = v^*)$$

$$\geq \quad \frac{\theta}{d+\theta}(1) + \frac{1}{d+\theta} \cdot \frac{1}{2} \cdot \left( 1 - 2\left(\frac{d}{d+\theta}\right)^{t-1} \right)$$

$$= \quad 1 - \frac{d}{2(\theta+d)} - \left(\frac{d}{d+\theta}\right)^t.$$

Since the ML estimator performs at least as well as the ball centrality estimator, the claim follows.

### C.4  Proof of theorem 4.1

To analyze the probability of detection under the first-timestamp estimator, we consider probability of the true source reporting before any other node, or $\mathbb{P}(\tau_0 < \tau_m)$. We use $R_i$ to denote the random time delay between the infection times of $i$'s parent and $i$; by "parent", we mean with respect to the infected subtree $G_t$, which is rooted at node $v^* = 0$. Moreover, we let the source have a different degree $d_0$ than the rest of tree for simplicity of calculation. We write the probability of detection explicitly in Figure 11. The expression starts by conditioning on the reporting time of the true source, $\tau_0$, then conditions on the times at which other nodes receive the message. Recall that node $i$ receives the message at time $X_i$, and reports the message to the adversary at time $\tau_i$.

Expression (25) can be written recursively. Define

$$g(a) \quad \triangleq \quad e^{-a(\lambda_2+d-2)} \int_0^a e^{u(\lambda_2+d-2)}[1 + g(a-u)]^{d-1} du.$$

$$\tag{26}$$

Letting $s = a - u$, we get

$$g(a) = \int_0^a e^{-s(\lambda_2 + d - 2)}[1 + g(s)]^{d-1}ds.$$

Substituting $g(a)$ into Equation (25) gives

$$\mathbb{P}(\tau_0 < \tau_m) = \int_0^\infty \lambda_2 e^{-t_0(\lambda_2 + d_0)}[1 + g(t_0)]^{d_0}dt_0, \tag{27}$$

where $d_0$ is the degree of the source. To make this expression symmetric with respect to the recursive function $g(\cdot)$, we assume that the source has degree $d - 2$, and all other nodes have degree $d$, so $d_0 = d - 2$. We subsequently compute the probability of detection by solving for $g(t_0)$, which can be written as a differential equation:

$$g'(a) = e^{-a(\lambda_2 + d - 2)}[1 + g(a)]^{d-1},$$

with initial condition $g(0) = 0$. This separable, nonlinear differential equation can be solved exactly, giving

$$g(a) = \left(\frac{(d-2)e^{-a(\lambda_2 + d - 2)} + \lambda_2}{\lambda_2 + d - 2}\right)^{-\frac{1}{d-2}} - 1.$$

Substituting into (27), we obtain the exact probability of detection:

$$\mathbb{P}(\tau_0 < \tau_m) = \frac{\lambda_2(\log(\lambda_2 + d - 2) - \log(\lambda_2))}{d - 2}.$$

Letting $\lambda_2 = \theta$, we get

$$\mathbb{P}(\tau_0 < \tau_m) = \frac{\theta}{d - 2}\log\left(\frac{d + \theta - 2}{\theta}\right), \tag{28}$$

which is the final expression.

### C.5 Proof of theorem 4.2

To prove this claim, we analyze the (suboptimal) reporting centrality estimator, which achieves the condition in Theorem 4.2. Our goal is to show that under an eavesdropper adversary, reporting centrality has a strictly positive probability of detection as $d \to \infty$.

Let $R_t = \{v \in V_t | R_v(t) = 1\}$ denote the set of reporting centers at time $t$. We compute the probability of detection by conditioning on the event $v^* \in R_t$. For brevity of notation, we use $\hat{v}$ to denote the reporting centrality source estimate in this proof. For any fixed time $t$, we have

$$\mathbb{P}(\mathtt{M_{RC}}(\boldsymbol{\tau}, G) = v^*) = \underbrace{\mathbb{P}(v^* \in R_t)}_{(a)} \times$$
$$\underbrace{\mathbb{P}(\mathtt{M_{RC}}(\boldsymbol{\tau}, G) = v^* | v^* \in R_t)}_{(b)}. \tag{29}$$

Note that the probability of detection is defined for a fixed $t$. Our goal is to lower-bound this quantity as $t \to \infty$. The proof consists of three steps:

(a) Show that $\liminf_{t \to \infty} \mathbb{P}(v^* \in R_t) \geq C_d > 0$.

(b) Show that $\mathbb{P}(\mathtt{M_{RC}}(\boldsymbol{\tau}, G) = v^* | v^* \in R_t) = 1$.

(c) Combine parts $(a)$ and $(b)$ to give the claim.

For readability, we abbreviate our notation in this proof. The number of nodes in the subtree $T_w^{v^*}$ will be denoted $N_w(t)$ instead of $N_{T_w^{v^*}}(t)$, and the number of *reporting* nodes in this subtree will be denoted $Y_w(t)$ instead of $Y_{T_w^{v^*}}(t)$. Thus $N_1(t)$ denotes the number of infected nodes in the first subtree of $v^*$, and $Y_1(t)$ denotes the number of reporting nodes in the first subtree of $v^*$.

**Part (a)**: Show that $\liminf_{t \to \infty} \mathbb{P}(v^* \in R_t) \geq C_d$.

We demonstrate this by conditioning on the event that $v^*$ is the unique *rumor* center in $G_t$. This happens if and only if $\forall w \in \mathcal{N}(v^*)$, $N_w(t) < N(t)/2$ [31], where $\mathcal{N}(v^*)$ denotes the neighbors of $v^*$. We let $C_t = \{v \in V_t \mid v = \text{rumor center of } G_t\}$, which gives

$$\mathbb{P}(v^* \in R_t) \geq \underbrace{\mathbb{P}(v^* \in C_t, |C_t| = 1)}_{(a_1)} \underbrace{\mathbb{P}(v^* \in R_t | v^* \in C_t, |C_t| = 1)}_{(a_2)}$$

Part $(a_1)$ is studied in Theorem 3.1 of [31], which shows that

$$\liminf_{t \to \infty} \quad \mathbb{P}(v^* \in C_t, |C_t| = 1)$$
$$= 1 - d\left(1 - I_{1/2}\left(\tfrac{1}{d-2}, 1 + \tfrac{1}{d-2}\right)\right), \tag{30}$$

where $I_{1/2}(a, b)$ is the regularized incomplete Beta function, or the probability that a Beta random variable with parameters $a$ and $b$ takes a value in $[0, 1/2)$.

For part $(a_2)$, we show that $\lim_{t \to \infty} \mathbb{P}(v^* \in R_t | v^* \in C_t, |C_t| = 1) = 1$. Our approach is to first show that the fraction of reporting nodes in each tree converges almost surely to a constant as $t \to \infty$. We subsequently show that if $v^*$ is a unique rumor center, it is almost surely a reporting center as $t \to \infty$.

**Lemma C.1** *For all $i \in [d]$, the following condition holds as $t \to \infty$:*

$$\frac{Y_i(t)}{N_i(t)} \xrightarrow{a.s.} \alpha_{d,\theta} = \frac{\theta}{d + \theta - 2}. \tag{31}$$

(Proof in Section C.5.1)

This lemma states that the ratio of reporting to infected nodes in each subtree converges almost surely to a constant. The proof proceeds by describing the evolution of each subtree and the adversary's observations as a generalized Pólya urn process with negative coefficients. The ratio of balls in this urn process can be shown to converge almost surely, which implies the claim. Now we use Lemma C.1 to show that $v^*$ is a reporting center.

Since $v^*$ is a unique rumor center, $\forall i \in [d]$ (where $[d] = \{1, 2, \ldots, d\}$), $N_i(t) < N(t)/2$ [30]. Because of this conditioning and the fact that the infected subtree sizes (normalized by the total number of infected nodes) converge almost surely [31], for any outcome $\omega$ of the underlying diffusion process, it holds that $N_i(t, \omega)/N(t, \omega) \xrightarrow{t \to \infty} (1/2 - \delta_i(\omega))$, for some $\delta_i(\omega) > 0$. Here $N_i(t, \omega)$ denotes the number of infected nodes in subtree $i$ at time $t$ under outcome $\omega$. We wish to show that for any given, feasible set of offsets $\delta_i(\omega)$, $i \in [d]$, $v^*$ eventually becomes a reporting center w.p. 1.

Since $Y_i(t)/N_i(t) \xrightarrow{a.s.} \alpha_{d,\theta}$, for a given outcome $\omega$ of the underlying process, we can write $Y_i(t, \omega) = N_i(t, \omega)(\alpha_{d,\theta} + \epsilon_i(t, \omega))$, where $\epsilon_i(t, \omega) \in \mathbb{R}$. Note that $\epsilon_i(t, \omega) \xrightarrow{t \to \infty} 0$. We write the condition for being a reporting center as

$$Y_i(t, \omega) \stackrel{?}{<} \frac{Y(t, \omega)}{2}$$
$$\implies N_i(t, \omega)(\alpha_{d,\theta} + \epsilon_i(t, \omega))$$
$$\stackrel{?}{<} \frac{\alpha_{d,\theta} N(t, \omega)}{2} + \frac{\sum_{j \in [d]} \epsilon_j(t, \omega) N_j(t, \omega)}{2}$$
$$\implies N(t, \omega)(\tfrac{1}{2} - \delta_i(\omega))(\alpha_{d,\theta} + \epsilon_i(t, \omega))$$
$$\stackrel{?}{<} \frac{1}{2}\left(\alpha_{d,\theta} N(t, \omega) + \sum_{j \in [d]} \epsilon_j(t, \omega) N(t, \omega)(\tfrac{1}{2} - \delta_j(\omega))\right)$$
$$\implies \epsilon_i(t, \omega) - \frac{\sum_j \epsilon_j(t, \omega)(\tfrac{1}{2} - \delta_j(\omega))}{1 - \frac{\delta_i(\omega)}{2}} \stackrel{?}{<} \alpha_{d,\theta} \cdot \delta_i(\omega). \tag{32}$$

Note that $\alpha_{d,\theta}$ and $\delta_i(\omega)$ are both strictly positive, and

$$\lim_{t\to\infty} \epsilon_i(t,\omega) - \frac{\sum_j \epsilon_j(t,\omega)(\frac{1}{2} - \delta_j(\omega))}{1 - \frac{\delta_i(\omega)}{2}} = 0.$$

Therefore, for every outcome $\omega$, there exists a time $T_\omega$ such that for all $t > T_\omega$, condition (32) is satisfied, so $\lim_{t\to\infty} \mathbb{P}(v^* \in R_t | v^* \in C_t, |C_t| = 1) = 1$.

Putting together parts $(a_1)$ and $(a_2)$, we get

$$\liminf_{t\to\infty} \mathbb{P}(v^* \in R_t)$$

$$\geq 1 - d\left(1 - I_{1/2}\left(\frac{1}{d-2}, 1 + \frac{1}{d-2}\right)\right).$$

**Part (b)**: Show that $\mathbb{P}(\mathtt{M_{RC}}(\tau, G) = v^* | v^* \in R_t) = 1$.

We show this by demonstrating that if $v^*$ is a reporting center, no other reporting centers exist.

We show that there can be at most one reporting center by contradiction. Suppose there are two nodes, $v^*$ and $w$, both of which are reporting centers. Consider the $d$ subtrees adjacent to $v^*$. Suppose the neighbors of $v^*$ are labelled $w_1, \ldots, w_d$. Each of these neighbors is the root of an infected subtree $T_{w_i}^{v^*}$. Since $v^*$ is a reporting center, we have two properties:

$$Y_{w_i}(t) < \frac{Y(t)}{2} \qquad \forall i \in [d] \tag{33}$$

$$Y_{v^*}(t) + \sum_{i\in[d]} Y_{w_i}(t) = Y(t). \tag{34}$$

Now suppose another node $u \neq v^*$ is also a reporting center. We label the neighbors of $u$ as $z_1, \ldots, z_d$. We will use This implies that

$$Y_{T_{z_i}^u}(t) < \frac{Y(t)}{2} \qquad \forall i \in [d] \tag{35}$$

$$Y_u(t) + \sum_{i\in[d]} Y_{T_{z_i}^u}(t) = Y(t). \tag{36}$$

Suppose without loss of generality that $z_1, w_1 \in P(u, v^*)$. In order to satisfy condition (35), it must hold that $Y_{v^*}(t) + \sum_{i=2}^d Y_{w_i}(t) < \frac{Y(t)}{2}$. Substituting this condition into (34), and using condition (33) implies that $Y_{w_1}(t) > \frac{Y(t)}{2}$. This is a contradiction because $v^*$ is assumed to be an element of $R_t$, which implies that $Y_{w_1}(t) < Y(t)/2$. Therefore, there can be no more than one reporting center, so $\mathbb{P}(\mathtt{M_{RC}}(\tau, G) = v^* | v^* \in R_t) = 1$.

**Part (c)**: Combine parts (a) and (b) to give the final claim.

We wish to lower bound $\liminf_{t\to\infty} \mathbb{P}(\mathtt{M_{RC}}(\tau, G) = v^*)$. Recall from (29) that $\mathbb{P}(\mathtt{M_{RC}}(\tau, G) = v^*) \geq \mathbb{P}(v^* \in R_t)\mathbb{P}(\mathtt{M_{RC}}(\tau, G) = v^* | v^* \in R_t)$. We have bounds for each term:

$$\liminf_{t\to\infty} \mathbb{P}(v^* \in R_t)$$

$$\geq 1 - d\left(1 - I_{1/2}\left(\frac{1}{d-2}, 1 + \frac{1}{d-2}\right)\right)$$

$$\mathbb{P}(\mathtt{M_{RC}}(\tau, G) = v^* | v^* \in R_t) = 1,$$

which gives

$$\liminf_{t\to\infty} \mathbb{P}(\hat{v} = v^*) \geq 1 - d\left(1 - I_{1/2}\left(\frac{1}{d-2}, 1 + \frac{1}{d-2}\right)\right),$$

thereby proving the claim.

### C.5.1 Proof of lemma C.1

The evolution of the (partially unobserved) infected subgraph on regular trees can be described by a Pólya urn process with $d$ colors of balls. Each ball represents one active edge between an infected and an uninfected node, and all active edges in the same source-adjacent subtree have the same color. At $t = 0$, we have one ball of each color, since there are $d$ active edges extending from the true source.

Because of the memorylessness of spreading, each of the active edges in $G_t$ is equally likely to spread the infection next. The urn evolves as follows: pick a ball uniformly at random; the subtree corresponding to the drawn color spreads the message over one of its active edges, infecting a neighbor $w$. Once $w$ is infected, one active edge is removed from the subtree (i.e., the edge that just spread the message), and $d - 1$ new active edges are added (from $w$ to its uninfected neighbors). In our urn, this corresponds to replacing the drawn ball and adding $d - 2$ balls of the same color. The replacement matrix for this urn can therefore be written as $A = (d - 2)I_d$, where $I_d$ denotes the $d \times d$ identity matrix.

This Pólya urn is well-studied, and in the limit, the fraction of balls of each color is known to converge to a Dirichlet distribution [18]. In order to model the eavesdropper's observations, we generalize the urn model by additionally giving each ball a pattern: striped or solid. Each solid ball corresponds to an active edge in the underlying diffusion process, whereas each striped ball represents an active edge from an infected node to the adversary. When such an edge fires, the adversary observes the timestamp of the sending node.

We adapt the previous urn dynamics in two key ways. First, when a solid ball is drawn, we still add $d - 2$ solid balls of the same color, but now we also add $\theta$ striped balls of the same color. These represent the $\theta$ independent connections between the node and the adversary. Second, when a striped ball is draw, we remove $\theta$ striped balls of the same color from the urn (i.e., the adversary only uses the first timestamp it receives). Thus, the replacement matrix for a single subtree looks like

$$A = \begin{array}{c} \\ \text{solid} \\ \text{stripe} \end{array} \begin{array}{c} \text{solid} \quad \text{stripe} \\ \begin{pmatrix} d - 2 & 0 \\ \theta & -\theta \end{pmatrix} \end{array}. \tag{37}$$

Let $s_n$ and $r_n$ denote the number of solid and striped balls, respectively, at the $n$th draw of the urn. The following condition holds as $n \to \infty$:

$$\frac{r_n}{s_n} \xrightarrow{a.s.} \frac{\theta}{d + \theta - 2}. \tag{38}$$

To show this, we use the following result from [13], simplified for clarity:

**Theorem 3.21 from [13]** *Consider a Pólya urn with replacement matrix A. Assume the following conditions:*

1. *For $i, j \in [d]$, $A_{ii} \geq -1$, and $A_{ij} \geq 0$ for $i \neq j$.*

2. *$A_{ij} < \infty$ for all $i, j \in [d]$.*

3. *The largest real eigenvalue $\lambda_A$ of A is positive, $\lambda_A > 0$.*

4. *The largest real eigenvalue $\lambda_A$ of A is simple.*

5. *The urn starts with at least one ball of a dominating type. A dominating type is a type of ball that, when drawn, produces balls of every other type.*

6. *$\lambda_A$ belongs to the dominating type.*

7. *The urn does not go extinct.*

*Then $n^{-1}[s_n \ \ r_n]^{\mathsf{T}} \xrightarrow{a.s.} \lambda_A v$, where $\mathsf{T}$ denotes the transpose of a vector, $\lambda_A$ is the largest eigenvalue of replacement matrix A, and $v$ is the corresponding right eigenvector.*

Conditions 1 and 2 are satisfied by examination of $A$. The eigenvalues of $A$ are $(d - 2)$ and $-\theta$, so conditions 3 and 4 are satisfied. Conditions 5 and 6 are met because $\lambda_A$ belongs to the class of solid

balls (i.e., a dominating type), and the urn starts with a solid ball by construction. Condition 7 is met because solid balls are never removed, and we start with one solid ball. Thus, Theorem 3.21 from [13] applies, which implies that $\frac{r_n}{s_n} \xrightarrow{a.s.} \frac{v_2}{v_1} = \frac{\theta}{d+\theta-2}$, since the eigenvector for $\lambda_A = (d-2)$ is $\boldsymbol{v} = [d+\theta-2, \theta]$.

There is a one-to-one mapping between the evolution of such an urn and the spreading of the message. Without loss of generality, we consider the evolution of the first subtree, $N_1(t)$. Let $\beta_n$ denote the time at which the $nth$ ball is drawn. We define $s(t) = \max_{\{n:\beta_n \leq t, \beta_{n+1} > t\}} s_n$ and $r(t) = \max_{\{n:\beta_n \leq t, \beta_{n+1} > t\}} r_n$. We can now map the number of reporting and infected nodes to the evolution of the Pólya urn:

$$
\begin{aligned}
s(t) &= 1 + (d-2)N_1(t) \\
r(t) &= N_1(t) - Y_1(t).
\end{aligned}
$$

Solving for $Y_1(t)$ and $N_1(t)$ and taking the limit gives
$\lim_{t\to\infty} \frac{Y_1(t)}{N_1(t)} = 1 - \frac{d-2}{\theta} \lim_{t\to\infty} \frac{r(t)}{s(t)-1} = \frac{\theta}{d+\theta-2}$ with probability 1, since both $s(t)$ and $r(t)$ tend to infinity as $t \to \infty$.

### C.6 Proof of corollary B.1

The outline of this proof is similar to that of Theorem 4.2. One difference is that in the spy-based adversary, the true source $v^*$ never reports to the adversary, so $Y_{v^*}(t) = 0$. This does not change the proof in any significant way. We again condition on the event that the true source $v^*$ is a reporting center, which gives.

$$
\mathbb{P}(\mathtt{M_{RC}}(\boldsymbol{\tau}, G) = v^*) = \underbrace{\mathbb{P}(v^* \in R_t)}_{(a)} \times
$$
$$
\underbrace{\mathbb{P}(\mathtt{M_{RC}}(\boldsymbol{\tau}, G) = v^* | v^* \in R_t)}_{(b)}. \tag{39}
$$

Recall that $R_t$ is the set of reporting centers at time $t$. For part (b), if $v^*$ is a reporting center, then it is a unique reporting center (from Proof C.5), so the probability of detection is 1. Thus, the key is to characterize (a), $\mathbb{P}(v^* \in R_t)$, as $t \to \infty$.

As before, we lower bound this quantity by conditioning on the event that $v^*$ is the unique *rumor center* in $G_t$. We let $C_t = \{v \in V_t \mid v = \text{rumor center of } G_t\}$, which gives

$$
\mathbb{P}(v^* \in R_t) \geq \underbrace{\mathbb{P}(v^* \in C_t, |C_t| = 1)}_{(a_1)} \underbrace{\mathbb{P}(v^* \in R_t | v^* \in C_t, |C_t| = 1)}_{(a_2)}
$$

We know from Proof C.5 that part $(a_1)$ gives

$$
\begin{aligned}
\liminf_{t\to\infty} \quad & \mathbb{P}(v^* \in C_t, |C_t| = 1) \\
= \quad & 1 - d\left(1 - I_{1/2}\left(\tfrac{1}{d-2}, 1 + \tfrac{1}{d-2}\right)\right),
\end{aligned} \tag{40}
$$

where $I_{1/2}(a, b)$ is the regularized incomplete Beta function, or the probability that a Beta random variable with parameters $a$ and $b$ takes a value in $[0, 1/2)$.

For part $(a_2)$, we show that $\lim_{t\to\infty} \mathbb{P}(v^* \in R_t | v^* \in C_t, |C_t| = 1) = 1$. This portion is the only real difference in proof between the present corollary and Theorem 4.2. Again, we show that the fraction of reporting nodes (or spies) in each tree converges almost surely to a constant as $t \to \infty$. However, unlike the eavesdropper adversary, the spy-based adversary does not require Pòlya urns to make this case.

**Claim C.2** *For all $i \in [d]$, the following condition holds as $t \to \infty$:*

$$
\frac{Y_i(t)}{N_i(t)} \xrightarrow{a.s.} p. \tag{41}
$$

This claim follows easily from the central limit theorem, since the number of spy nodes (or reporting nodes) in the $i$th subtree is simply a Binomial$(N_i(t), p)$ random variable. Recall that $N_i(t)$ denotes the number of infected nodes in the $i$th subtree adjacent to $v^*$. Therefore, by the same argument as Proof C.5, if $v^*$ is a unique rumor center, it is almost surely also a reporting center as $t \to \infty$.

This, in turn, implies the overall result:

$$\liminf_{t \to \infty} \mathbb{P}(\texttt{M}_{\texttt{RC}}(\boldsymbol{\tau}, G) = v^*) \geq C_d > 0.$$

where

$$C_d = 1 - d \left( 1 - I_{1/2} \left( \frac{1}{d-2}, 1 + \frac{1}{d-2} \right) \right).$$