[Reviews · NeurIPS 2017]

Reviewer 1



The paper explores the effect of modifying the transaction-flooding protocol of the bitcoin network as a response to deanonymisation attacks in the original pre-2015 protocol. The authors model the original protocol (trickle) and the revised variant (diffusion) and suggest that deanonymisation can equally be established with both protocols. The authors further support their claim by exploring both models in simulations. Quality of Presentation: The paper is well-written (minor aspects: some incomplete sentences, unresolved reference) and provides the reader with the necessary background information to follow. The representation of the transaction-flooding protocols is comprehensive and introduced assumptions are highlighted. Content-related questions: - The authors refer to probability of detecting participants of 30%, which is relatively low (but indeed considerable). A fundamental assumption of performing the deanonymisation, however, is that the listening supernode would connect to all active bitcoin nodes. How realistic would it be in the future (assuming further growth of the network) that a supernode would be able to do so? -- Edit: The argument for the use of botnets made by the authors is a convincing counterpoint. - While the overall work is involved, I find that the paper does not comprehensively answer the original claim that both trickle and diffusion perform equally good or bad at maintaining anonymity. Observing Figure 6, we can find that Trickle and Diffusion simulated show differences in probability of around 0.1 (with Diffusion performing worse than Trickle), which is not negligible. Figure 7 shows the operation on a snapshot of the real network, with an inverted observation: Diffusion performs better than Trickle. (Admittedly, they converge for larger number of eavesdropper connections.) The discrepancy and conflicting observations are worth discussing, and the statement that both protocols have `similar probabilities of detection' needs to be clarified (How do you establish 'similarity'?).

Reviewer 2



The paper is extremely interesting, very well motivated, and was a pleasure to read! 1.] The paper is extremely well written and overall very clear. 2.] Can comparisons be made with existing methods? If so, why weren't these made? 3.] Can the authors discuss robustness to model misspecification in detail (both strengths and weakness)? 4.] Can the authors discuss the sensitivity of any fixed tuning parameters in the model (both strengths and weakness)? 5.] What is the scalability of the model proposed and computational complexity? Will the authors be making the code publicly available with the data? Are all results reproducible using the code and data? 6.] What conclusion should a user learn and drawn? The applications section was a bit disappointing given the motivation of the paper. A longer discussion is important to the impact and success of this paper. Please discuss.

Reviewer 3



The paper presents an interesting approach towards deanonymizing the Bitcoin network. The paper is presented in a tidy manner and contains enough figures to explain the approach and experiments. Especially the probabilistic analysis is nice and provides stimulating ideas. I did not find any problems and I recommend acceptance. Though the paper is well presented, I'm not completely sure that the paper fits to NIPS or could have stimulating impact to the NIPS community